# Learning Enhanced Representations for Tabular Data via Neighborhood Propagation

**Kounianhua Du**[†]**, Weinan Zhang**[‡]**, Ruiwen Zhou**[†]**, Yangkun Wang**[†]**, Xilong Zhao**[†]**, Jiarui Jin**[†]

Department of Computer Science
Shanghai Jiao Tong University
{774581965, wnzhang, skyriver, espylacopa, zhaoxilong, jinjiarui97}@sjtu.edu.cn

**Quan Gan, Zheng Zhang, David Wipf**
Amazon
{quagan, zhaz, daviwipf}@amazon.com

## Abstract

Prediction over tabular data is an essential and fundamental problem in many important downstream tasks. However, existing methods either treat a data instance of the table independently as input or do not jointly utilize multi-row features and labels to directly change and enhance target data representations. In this paper, we propose to 1) construct a hypergraph from relevant data instance retrieval to model the cross-row and cross-column patterns of those instances, and 2) perform message **P**ropagation to **E**nhance the target data instance representations for **T**abular prediction tasks. Specifically, our tailored message propagation step benefits from both the fusion of label and features during propagation, as well as locality-aware high-order feature interactions. Experiments on two important tabular data prediction tasks validate the superiority of the proposed PET model relative to other baselines. Additionally, we demonstrate the effectiveness of the model components and the feature enhancement ability of PET via various ablation studies and visualizations. The code is available at https://github.com/KounianhuaDu/PET.

## 1 Introduction

Prediction over tabular data is a fundamental and essential problem in many data science applications including recommender systems (Bobadilla et al., 2013; Ying et al., 2018), online advertising (Richardson et al., 2007; Zhou et al., 2018), fraud detection (Bolton and Hand, 2002), question answering (Chen et al., 2020), etc. Most existing methods seek to capture the patterns of feature interactions within an instance independently using tree models (Chen and Guestrin, 2016) or deep networks (Guo et al., 2017).

Recently, as shown by Papernot and McDaniel (2018), feeding in neighbors of the target as input can improve the robustness of data representations and help the model generalize better to out-of-distribution samples. Some retrieval based methods (Pi et al., 2020; Qin et al., 2020, 2021) then seek to utilize multiple neighboring data instances of the target for label prediction. These methods take the auxiliary instances as extra inputs but do not consider that such auxiliary information could enhance the target representation. Other existing graph-based methods on tabular data (You et al., 2020; Wu et al., 2021; Guo et al., 2021b) aim to learn robust representations with locality structure, as Verma and Zhang (2019) proved the stabilization and generalization ability of graph neural networks.

---

[†]Work done during internship at Amazon Web Services Shanghai AI Lab.
[‡]Weinan Zhang is the corresponding author.

36th Conference on Neural Information Processing Systems (NeurIPS 2022).

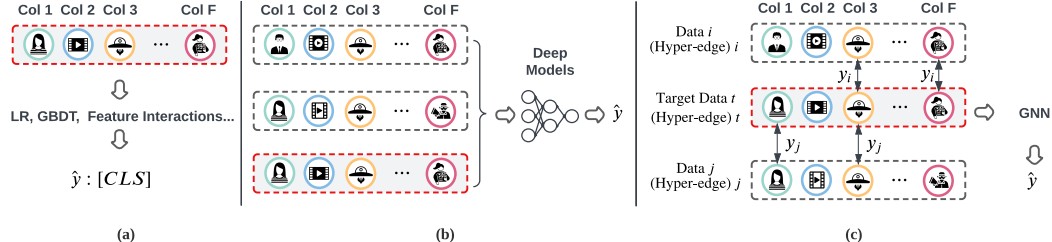

Figure 1: Difference between PET and other models. (a) The popular tree models and interaction-based models utilize a single data instance for prediction. (b) The retrieval-based methods take multiple data instances as input without sufficiently mining the interaction patterns among them. (c) The proposed PET models the multiple data instances set as a hypergraph and capture their correlations with the assistance of labels.

However, they either omit the high-order product feature interactions, or else serve as a plugin while resorting to other architectures like factorization machines (Rendle, 2010) to explore feature interactions. Moreover, they ignore the mutual enhancement between the representations of labels and features. In this paper, with the aim of learning enhanced data representations of tabular data, we propose to construct a hypergraph among relevant data instances to model the set relationships (Srinivasan et al., 2021). Additionally, we design an end-to-end graph neural network prediction model that generates high-order feature interactions with the assistance of the locality structure and label adjustment.

Specifically, we design PET, a novel architecture that **P**ropagates and **E**nhances the **T**abular data representations based on the hypergraph for target label prediction. We first retrieve from the observed data instances pool to get the neighboring data instances for each target. The resulting instance set can be seen as a hypergraph, where sets of feature columns of data instances form hyperedges and each distinct feature value of these data instances form a node. This hypergraph models the cross-row and cross-column relations of the resulting instance set. Then we conduct message propagation on the graph. The propagation serves three purposes. First, auxiliary label information from the retrieved data instances propagates through the common feature value nodes to help the target prediction. Second, feature representations get enhanced through the locality structure. Our interactive message generation, attention-based aggregation, and update generate locality-aware high-order feature interactions. Third, the labels are incorporated into the propagating messages to directly adjust the feature spaces and generate label-enhanced feature representations.

The main contributions are summarized as follows:

- We propose a retrieval-based hypergraph to capture the feature and label correlations among tabular data instances.
- We design an end-to-end graph neural network prediction model that unifies the product feature interaction, locality mining, and label enhancement.
- We utilize the observed labels in the resulting set to guide the feature learning process and use the propagated labels to enhance predictions.

We evaluate the proposed PET model on two prediction tasks, i.e., binary classification and top-n ranking, over five tabular datasets, where substantial performance improvement against other strong baselines validates the superiority of PET.

## 2   Preliminaries and Related Work

**Tabular data prediction.** Tabular data prediction treats every row of the table as a data instance and every column/field as a feature attribute.[1] We consider tabular data prediction under a single table scenario in this paper, and prediction over multiple tables can be seen as the prediction over a joined table. We also focus on tabular *discrete* data in this paper, while the continuous feature values can be

---

[1]We use *row* and *data instance*, as well as *column* and *field* interchangeably.

discretized in various ways (Guo et al., 2021a). Currently some of the most widely used models for tabular data prediction include Gradient Boosting Decision Trees (GBDTs) (Friedman, 2001) and Factorization Machines (FM) (Rendle, 2010). Variants of FM such as DeepFM (Guo et al., 2017) and Wide & Deep (Cheng et al., 2016) are especially popular in industrial applications. For a given row, such models simply take in the row and make predictions, i.e., these models assume that the rows are IID. Such an assumption, however, is reasonable only if the embedding representations of the tabular data can sufficiently host the high-order interaction patterns within the data instance, which is practically impossible.

**Graph neural networks on tabular data.** As Graph Neural Networks (GNNs) become popular, multiple attempts to apply GNNs on tabular data prediction have emerged. Since the model takes in a graph that connects the rows and columns together, the prediction on a single row no longer depends only on itself, but also other rows, and is therefore suitable for exploiting non-IID properties in tabular data. Examples of using GNNs on tabular data include Wu et al. (2021) that constructs a mini-batch data-feature bipartite graph, You et al. (2020) that treats the (incomplete) table as an adjacency matrix of a bipartite graph, and Guo et al. (2021b) that treats rows as nodes and build edges according to predefined rules. These methods omit the product feature interactions or else resort to other architecture like factorization machines to explore them. Moreover, they ignore the mutual enhancement between labels and features.

**Hypergraphs and hyperedge classification/regression.** If we treat each individual value in a table as a single node, then a row describes an $n$-ary relationship among the nodes. Normal graphs have trouble describing this relation since edges can only connect two nodes at a time. A *hypergraph* generalizes graphs such that an edge can connect more than two nodes. It is defined as a pair $G = (V, E)$ with its node set $V$ and its *hyperedge* set $E \subseteq \mathcal{P}(V)$, where $\mathcal{P}(V)$ is the power set of $V$, meaning that each "edge" becomes simply a subset of $V$, regardless of the number of nodes in the "edge". We can construct a hypergraph from a set of rows, where each hyperedge correspond to a row and each node correspond to the distinct feature values among all the rows. Tabular data prediction problem can be cast into a hyperedge classification/regression problem, where every row now corresponds to a hyperedge.

**Hypergraph neural networks.** Hypergraph neural networks are an adaptation of GNNs with a message passing paradigm whereby node representations are used to update hyperedge representations, which in turn update the node representations again (Srinivasan et al., 2021; Bai et al., 2021; Feng et al., 2019). Using a hypergraph neural network on tabular data allows us to obtain a representation for each row from its corresponding hyperedge, as well as a representation for each individual value of each column from its corresponding node.

# 3   Methodology

When making prediction for a sample, feeding in similar or relevant samples together as input is known to contribute to robustness and generalization abilities for out-of-distribution samples (Papernot and McDaniel, 2018). In light of this, for a given data instance, our model retrieves a set of relevant data instances according to a relevance metric, aiming to take auxiliary information from relevant data instances.

The resulting instances set can be seen as a hypergraph, where each distinct feature value forms a node and a collection of them, i.e., a data instance, forms a hyperedge. After the star expansion (Agarwal et al., 2006; Srinivasan et al., 2021), we get a bipartite graph with feature value nodes on one side and data instance nodes on the other, on which we propagate and enhance representations. An advantage of message passing is that it allows us to capture higher-order interactions among nodes and hyperedges (Srinivasan et al., 2021), i.e. the interactions among the individual column values as well as the rows. The other advantage of message passing is that we can utilize the labels of retrieved data instances to interact with features to generate label-enhanced messages, guide the message aggregation process, and take advantage of label propagation at the same time. After message passing, the enhanced data instance representations are then used for prediction.

Figure 2 illustrates the framework of PET. Detailed descriptions of each individual component are provided in the following subsections.

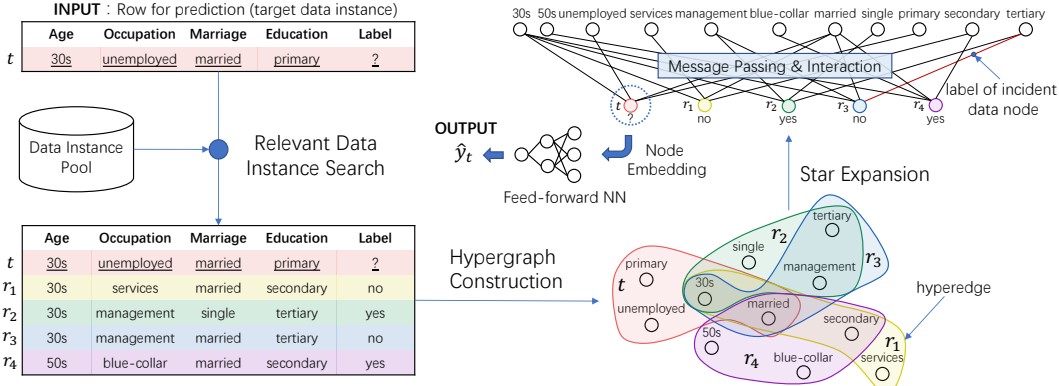

Figure 2: The workflow of PET. For each row to predict (top-left), PET retrieves a fixed number of relevant data instances (bottom-left) and constructs a hypergraph (bottom-right) from the resulting data instances set. After a star expansion (top-right), we get a data-feature bipartite graph with data instance nodes at the bottom and feature value nodes at the top. We then perform the proposed hypergraph neural network on the resulting graph and use the target data instance node representation for prediction.

## 3.1 Graph Construction

For each target data instance, we retrieve $K$ relevant data instances from the observed data instances set using ElasticSearch[2] and construct a hypergraph from the resulting instance set.

Let $X^t = \{x_f^t | f = 1, \ldots, F\}$ be the feature of a target data instance $t$, where $F$ is the number of feature fields and $x_f^t$ is the feature value of the $f$-th field of data instance $t$. We first conduct a boolean query to obtain instances that have at least one common feature value with the target data instance. Then we retrieve the top-$K$ relevant instances $r_1, \cdots, r_K$ from the filtered instances set according to the relevance value (Qin et al., 2021) defined as:

$$R(X^t, X^j) = \sum_{f=1}^{F} \text{IDF}(x_f^t) \cdot 1_{(x_f^t = x_f^j)}, \tag{1}$$

$$\text{IDF}(x_f^t) = \log \frac{N - N(x_f^t) + 0.5}{N(x_f^t) + 0.5}, \tag{2}$$

where $1_{(\cdot)}$ is the indicator function, $N$ is the number of data instances in the table, and $N(x_f^t)$ is the number of data instances that have feature value $x_f^t$ in the $f$-th field. This metric is equivalent to the BM25 (Robertson et al., 1995) metric if we treat each data instance as a document and their feature values as the terms.

After the retrieval, the resulting instances set $\{X^t, X^{r_1}, \cdots, X^{r_K}\}$ can be seen as a hypergraph, where each distinct feature value in each individual field forms a node and each data instance constructs a hyperedge. We then perform a star expansion (Agarwal et al., 2006) on the hypergraph, i.e. construct a bipartite graph $G = (V_D, V_F, E)$ where $V_D = \{t, r_1, \ldots, r_K\}$, $V_F = \{x_f^j | f = 1, \cdots, F; j = t, r_1, \ldots, r_K\}$, and an undirected edge exists between two nodes $i \in V_F$ and $j \in V_D$ if $i \in X^j$.

## 3.2 Message Passing and Interaction

After the graph is constructed, we propagate and enhance the representations on it. The propagation serves three purposes. First, the label information propagates through common feature value nodes to help the label prediction of the target data instance node. Second, the features get enhanced through taking in high-order information. Locality-aware high-order product feature interactions are generated through the interactive message generation and attention based aggregation. Third,

[2]https://www.elastic.co/elasticsearch/

the label embeddings directly interact with the features to adjust the feature spaces and generate label-enhanced features and high-order interactions. Then we detail the components as follows.

**Initialization.** Before message passing, we initialize the node representations and edge representations with trainable embedding vectors. Let $\Phi_x$ and $\Phi_y$ be different trainable embedding layers for feature value nodes and data instances nodes, respectively. And let $\Phi_{in}$ and $\Phi_{out}$ be the embedding layers for the edges. We first initialize the feature value node representations as

$$h_i^{(0)} = \Phi_x(i), i \in V_F. \tag{3}$$

We also initialize the data instance node representations of the retrieved rows with a trainable embedding vector associated with their labels. As for the target data node, we initialize its representations with a constant zero vector.

$$h_j^{(0)} = \begin{cases} \mathbf{0}, & j = t, \\ \Phi_y(y^j), & j \in V_D \backslash \{t\}, \end{cases} \tag{4}$$

where $y^j$ denotes the label of the data instance $j$.

In addition, we initialize each edge representation with the embedding of its incident data instance node label to guide the feature learning process and generate high-order interactions.

$$e_{ij}^{(0)} = \begin{cases} \Phi_{in}(y^i), & i \in V_D, j \in V_F, \\ \Phi_{out}(y^j), & i \in V_F, j \in V_D. \end{cases} \tag{5}$$

**Message Generation.** Then we use the interactions between edge and node representations along with the original node representations to generate the messages:

$$m_{ij}^{(l)} = (e_{ij}^{(l-1)} \odot h_i^{(l-1)}) \| h_i^{(l-1)}, \tag{6}$$

where $\odot$ denotes the Hadamard product, superscript $l$ means the $l$-th layer, and $\|$ denotes the concatenation operation.

As edge representations carry the label information initially, the Hadamard product term generates label-enhanced features. In addition, the edges will contain node features after updating edge representations with incident node representations, then the Hadamard product term produces high-order product feature interactions. The high-order product feature interactions prove to be important in many interaction-based tabular data prediction methods (Qu et al., 2018), which are often missed in other graph-based tabular networks or captured by an extra factorization machine (Wu et al., 2021).

**Message Aggregation.** The incoming neighboring messages are then aggregated based on an attention mechanism:

$$\begin{aligned} Q_j^{(l)} &= W_Q^{(l)} h_j^{(l-1)}, \\ K_{ij}^{(l)} &= W_K^{(l)} m_{ij}^{(l)}, \\ V_{ij}^{(l)} &= W_V^{(l)} m_{ij}^{(l)}, \\ a_{ij}^{(l)} &= \text{softmax}_{i \in N(j)}(Q_j^{(l)} K_{ij}^{(l)}), \\ n_j^{(l)} &= \sum_{i \in N(j)} a_{ij}^{(l)} V_{ij}^{(l)}. \end{aligned} \tag{7}$$

**Node Embedding Update.** After receiving the aggregated neighboring messgages, the node embeddings are updated based on the aggregated messages and their own node embeddings.

$$h_j^{(l)} = \sigma(W_N^{(l)}(h_j^{(l-1)} \| n_j^{(l)})), \tag{8}$$

where $\sigma(\cdot)$ denotes the ReLU activation function.

**Edge Embedding Update.** Then we update edge embeddings using their incident node embeddings as Equation (9). As edge embeddings are updated using the node embeddings, the features and labels on nodes are propagated to the edges as well. After rounds of propagation, the messages generated contain high-order feature interactions and high-order feature-label interactions.

$$e_{ij}^{(l)} = \sigma(W_E^{(l)}(h_i^{(l)} \| h_j^{(l)} \| e_{ij}^{(l-1)})). \tag{9}$$

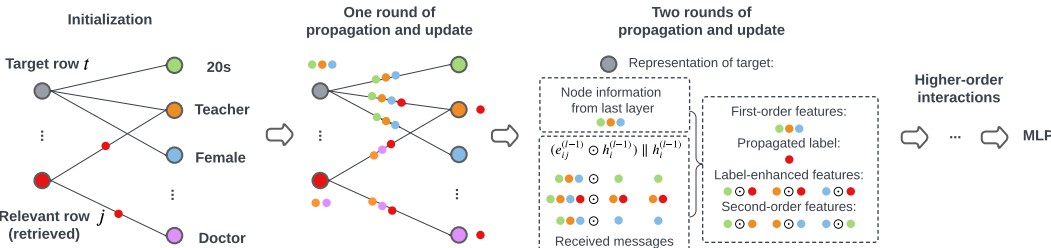

Figure 3: The representation of the target node after propagation. We use the red color to represent the retrieved instance node and use the grey color to represent the target data instance node to predict. Other feature value nodes are in different colors. The information flow is marked with the corresponding color. The dots-on-edges represent the feature and label information stored in edges. (best viewed in colors)

### 3.3 Label Prediction

Then we use the target data node embedding of the last layer for prediction:

$$\hat{y}_t = \text{MLP}(h_t^{(L)}), \tag{10}$$

For binary classification, the training objective is set as cross entropy:

$$\mathcal{L}(Y, \hat{Y}) = -\sum_t \left( y_t \log \hat{y}_t + (1 - y_t) \log(1 - \hat{y}_t) \right), \tag{11}$$

where $y_t$ is the label of the target data instance $t$.

### 3.4 Discussion

In this section, we discuss the strength and potential weakness of the proposed model and offer the complexity analysis for the proposed method.

#### 3.4.1 Model Analysis

We illustrate the representation of the target node after propagation in Figure 3. Following the proposed propagation method, the target node will receive the first order features, second-order features, label-enhanced features, and the propagated labels after two rounds of propagation. Propagating more than two layers will further produce higher-order feature interactions, cross-rows feature interactions, and feature-label interactions. Furthermore, the feature messages are adjusted by corresponding labels and attentively aggregated based on the locality structure.

When dealing with datasets with a large number of fields, the constructed graph may be dense. In this way, we could potentially meet the oversmoothing problem, which may degrade the model performance. However, this problem can still be overcome by some methods. For example, we could retrieve and construct the graph over selected fields (e.g., top 10 fields that have the largest mutual information with the labels).

#### 3.4.2 Complexity Analysis

In this section, we offer the complexity analysis for PET. Let $K$ indicate the number of retrieved data instances, $F$ indicate the number of feature fields, $N$ be the number of features, and $d$ be the embedding size.

For the retrieval stage, as discussed in RIM (Qin et al., 2021), the average length of the posting lists in the inverted index is $\frac{|SearchPool|}{N}$. The total time complexity of retrieval is $O(F\frac{|SearchPool|}{N})$. As for constructing the inverted index, it could be done offline and only once. The complexity of it is $O(F|SearchPool|)$.

The main computation of PET lies in the message passing part. The number of nodes in the graph for a data instance is no more than $(K + 1)(F + 1)$, and the number of edges is $2(K + 1)F$. Both of

them are $O(KF)$. The time complexity of single-layer message passing of PET is $O(KFd^2)$. This makes PET scalable as the complexity is linear with respect to the number of feature fields (number of table columns). In addition, the graph construction and message passing can be highly parallel: 1) The graph construction for each data instance can be done in parallel for a batch data. We batch all the resulting graphs for the data instances in the batch to form one single batched graph. Then we can apply GNN on the resulting single batched graph to make predictions parallelly. 2) For the GNN, the operation of the attention computation can be parallelized across all edges, and the computation of representations can be parallelized across all nodes and edges.

For the MLP computation, PET inputs the target node embedding for prediction. The complexity of it is $O(d^2)$, which is independent of the number of feature fields. In addition, the number of model parameters of PET is light. Both other methods (e.g., DeepFM, RIM, etc.) and PET have embedding tables of the same size $Nd$. The main differences lie in the linear layers of models. And for PET, the number of parameters of linear layers are $O(d^2)$, which is independent of the number of fields.

## 4 Experiments

In this section, we show the experimental results and the corresponding settings. Generally, we experiment on two kinds of tabular data prediction tasks: click-through rate (CTR) prediction and top-n recommendation. The results showcase the value of our model on tabular data prediction tasks. Additionally, we conduct several ablation studies to validate the components of our model.

### 4.1 Setup

We evaluate the performance of PET on five datasets. For the CTR prediction task, we conduct experiments on three large-scale datasets, i.e., Tmall[3], Taobao[4], and Alipay[5]. For the top-n recommendation task, we experiment on two widely-used public recommendation datasets, i.e., Movielens-1M[6] and LastFM[7]. The statistics of the used datasets are summarized in Table 1.

Table 1: Dataset statistics.

| Datasets | Samples | Fields |
|---|---|---|
| Tmall | 54,925,331 | 9 |
| Taobao | 100,150,807 | 4 |
| Alipay | 35,179,371 | 6 |
| Movielens-1M | 1,000,209 | 7 |
| LastFM | 18,993,371 | 5 |

The evaluation metrics include area under ROC curve (AUC) and negative log-likelihood (LogLoss) for the CTR prediction task and hit rate (HR), normalized discounted cumulative gain (NDCG), and mean reciprocal rank (MRR) for the top-n recommendation task.

Following Qin et al. (2021), we spilt the datasets according to the global timestamps. The earliest data instances are grouped into the retrieval pool. The latest data instances form the test pool. Then the remaining data instances are grouped into the train pool. To avoid unfair comparisons, the non-retrieval model takes the retrieval pool as an additional train pool.

On the CTR prediction task, we compare our model against nine widely-used and strong baselines. GBDT (Chen and Guestrin, 2016) is the widely-used tree model. DeepFM (Guo et al., 2017) is the inner-product interaction-based model. FATE (Wu et al., 2021) and TabGNN (Guo et al., 2021b) are the recent graph models. DIN (Zhou et al., 2018) and DIEN (Zhou et al., 2019) are the attention-based sequential models. SIM (Pi et al., 2020), UBR (Qin et al., 2020), and RIM (Qin et al., 2021) are the retrieval-based models. On the top-n recommendation task, we compare PET with six strong recommendation models, including factorization-based FPMC (Rendle et al., 2010) and TransRec (He et al., 2017), and recently proposed DNN models NARM (Li et al., 2017), GRU4Rec (Hidasi et al., 2016), SASRec (Kang and McAuley, 2018), and RIM (Qin et al., 2021).

As for the hyperparameters, we test the number of GNN layers in $\{2, 3\}$. The embedding sizes of all the models are consistent to ensure the fair comparison. More detailed hyperparameters and experiment settings are provided in Appendix A.5.

---

[3] https://tianchi.aliyun.com/dataset/dataDetail?dataId=42

[4] https://tianchi.aliyun.com/dataset/dataDetail?dataId=649

[5] https://tianchi.aliyun.com/dataset/dataDetail?dataId=53

[6] https://grouplens.org/datasets/movielens/1m/

[7] http://ocelma.net/MusicRecommendationDataset/lastfm-1K.html

Table 2: Result comparisons with baselines on CTR prediction task. ($K = 10$)

| Models | Tmall | | | Taobao | | | Alipay | | |
|---|---|---|---|---|---|---|---|---|---|
| | AUC | LogLoss | Rel.Impr. | AUC | LogLoss | Rel.Impr. | AUC | LogLoss | Rel.Impr. |
| GBDT | 0.8319 | 0.5103 | 12.08% | 0.6134 | 0.6797 | 44.08% | 0.6747 | 0.9062 | 32.36% |
| DeepFM | 0.8581 | 0.4695 | 8.66% | 0.6710 | 0.6497 | 31.71% | 0.6971 | 0.6271 | 28.1% |
| FATE | 0.8553 | 0.4737 | 9.01% | 0.6762 | 0.6497 | 30.70% | 0.7356 | 0.6199 | 21.40% |
| TabGNN | 0.8945 | 0.4158 | 4.24% | 0.7294 | 0.6173 | 21.17% | 0.8086 | 0.5849 | 10.44% |
| DIN | 0.8796 | 0.4292 | 6.00% | 0.7433 | 0.6086 | 18.90% | 0.7647 | 0.6044 | 16.78% |
| DIEN | 0.8838 | 0.4445 | 5.50% | 0.7506 | 0.6084 | 17.74% | 0.7502 | 0.6151 | 19.03% |
| SIM | 0.8857 | 0.4520 | 5.27% | 0.7825 | 0.5795 | 12.95% | 0.7600 | 0.6089 | 17.50% |
| UBR | 0.8975 | 0.4368 | 3.89% | 0.8169 | 0.5432 | 8.19% | 0.7952 | 0.5747 | 12.30% |
| RIM | 0.9138 | 0.3804 | 2.04% | 0.8563 | 0.4644 | 3.21% | 0.8006 | 0.5615 | 11.54% |
| **PET** | **0.9324** | **0.3321** | – | **0.8838** | **0.4162** | – | **0.8930** | **0.4132** | – |

Table 3: Result comparisons with baselines on top-n recommendation task. ($K = 10$)

| Datasets | Metric | FPMC | TransRec | NARM | GRU4Rec | SASRec | RIM | PET |
|---|---|---|---|---|---|---|---|---|
| ML-1M | HR@1 | 0.0261 | 0.0275 | 0.0337 | 0.0369 | 0.0392 | 0.0645 | **0.0904** |
| | HR@5 | 0.1334 | 0.1375 | 0.1418 | 0.1395 | 0.1588 | 0.2515 | **0.2889** |
| | HR@10 | 0.2577 | 0.2659 | 0.2631 | 0.2624 | 0.2709 | 0.4014 | **0.4404** |
| | NDCG@5 | 0.0788 | 0.0808 | 0.0866 | 0.0872 | 0.0981 | 0.1577 | **0.1903** |
| | NDCG@10 | 0.1184 | 0.1217 | 0.1254 | 0.1265 | 0.1341 | 0.2059 | **0.2390** |
| | MRR | 0.1041 | 0.1078 | 0.1113 | 0.1135 | 0.1193 | 0.1704 | **0.2006** |
| LastFM | HR@1 | 0.0148 | 0.0563 | 0.0423 | 0.0658 | 0.0584 | 0.0915 | **0.1149** |
| | HR@5 | 0.0733 | 0.1725 | 0.1394 | 0.1785 | 0.1729 | 0.3468 | **0.3621** |
| | HR@10 | 0.1531 | 0.2628 | 0.2227 | 0.2581 | 0.2499 | 0.5780 | **0.6033** |
| | NDCG@5 | 0.0432 | 0.1148 | 0.0916 | 0.1229 | 0.1163 | 0.2165 | **0.2381** |
| | NDCG@10 | 0.0685 | 0.1441 | 0.1185 | 0.1486 | 0.1409 | 0.2911 | **0.3156** |
| | MRR | 0.0694 | 0.1303 | 0.1083 | 0.1362 | 0.1289 | 0.2210 | **0.2492** |

## 4.2 Overall performance comparison

We first validate the effectiveness of the proposed PET model. The main results are summarized in Tables 2 and 3, where we can see the proposed PET performs consistently better on all datasets.

The results demonstrate the superiority of PET against the baselines on both tasks. On the CTR prediction task, PET achieves relatively $2.04\%$, $3.21\%$, $10.44\%$ higher AUC over the best performed baseline on Tmall, Taobao, Alipay, respectively. The results show that PET can learn effective tabular data representation for better prediction performance. Furthermore, compared with RIM that takes exactly the same inputs with PET, the improvements of PET are statistically significant under $95\%$ confidence level. Given that RIM also utilizes the labels of relevant rows, this empirically justifies the capability of our propagation method. On the top-n recommendation task, PET shows significant improvements on the recommendation task against other baselines, too. The results show that PET performs substantially better than the strong baselines in all comparisons.

## 4.3 Ablation study

### 4.3.1 Impact of the label usages

We further study the usage of labels of PET. The results are summarized in Table 4. For fast exploration, we randomly sample $10\%$ data from training pool and test pool on the three large CTR datasets, respectively. The best performed baseline RIM is tested on the sampled data for comparison. Both RIM and PET utilize the same set of retrieved data instances and their label information. We then examine the impact of different label usages on the sampled data. In the PET model, we initialize the embeddings of data instances nodes and edges with the corresponding label embeddings. We then inspect the impacts of such initialization and the operations on edges. For convenience, we calculate the homophily ratios for the resulting instances set. The homophily ratios for Tmall, Taobao, and Alipay are 0.5566, 0.6111, and 0.5097, respectively.

Table 4: Impact of label embeddings. (On randomly sampled data, $K = 10$)

| Models | Tmall | | Taobao | | Alipay | |
|---|---|---|---|---|---|---|
| | AUC | LogLoss | AUC | LogLoss | AUC | LogLoss |
| RIM | 0.9120 | 0.3769 | 0.8587 | 0.4586 | 0.7845 | 0.5742 |
| PET | 0.9279 | 0.3387 | 0.8762 | 0.4279 | 0.8720 | 0.4201 |
| PET (w/o edge labels) | 0.9291 | 0.3367 | 0.8665 | 0.4465 | 0.8558 | 0.4776 |
| PET (w/o node labels) | 0.9233 | 0.3494 | 0.8431 | 0.4847 | 0.8518 | 0.4799 |
| PET (w/o all labels) | 0.9208 | 0.3568 | 0.8416 | 0.4815 | 0.8096 | 0.5719 |

We first remove edge embedding initialization with labels and eliminate all the operations involved with edges, thus the message becomes

$$m_{ij}^{(l)} = h_i^{(l-1)}. \tag{12}$$

In addition, the edge embedding update step in Equation (9) is also omitted. As such, the labels only serve from propagation among nodes. The high-order product interaction between features and label-feature interaction are omitted. As shown in Table 4, PET without edge labels performs weaker than the original PET model, which demonstrates the power of high-order product interactions and label-enhanced features. Moreover, PET without edge labels still performs better than RIM, which validates the superiority of label propagation through common feature nodes against simple attention-based aggregation.

By removing the node labels, the embeddings of all the data instance nodes are initialized as constant vectors. In this way, no pure first-order label embeddings are used for prediction. Since any information related with labels will be propagated to nodes after a product interaction with features. One can see that PET without node labels performs better than RIM, which further justifies the effectiveness of high-order product interactions and label-enhanced features. Additionally, PET without node labels performs weaker than the original PET, which implies the power of initializing node embeddings with pure label embeddings for propagation.

We also provide PET (w/o all labels) to see the feature interactions under the framework. Concretely, we randomly initialize the label values. The results show the power of feature interactions and the advantages of feeding in labels.

### 4.3.2 Visualization of the feature distribution

In order to further explore the representation enhancement of PET, we visualize the features of PET and RIM with t-SNE (Van der Maaten and Hinton, 2008). Figure 4 illustrates the distributions of data instance embeddings and the feature embeddings of PET and RIM on Tmall. For both models, we randomly choose 10,000 samples from the train data and 10,000 samples from the test data. For PET, the data instance embeddings are taken from the data instance nodes (inputs of the MLP predictor) and the feature embeddings are the concatenated feature node embeddings for each data instance. For RIM, the data instance embeddings are the inputs of the final MLP predictor and the feature embeddings are the concatenated feature embeddings of each data instance.

From Figure 4, we can see that the distributions of train positive data points and train negative data points are more dissimilar than those of RIM. The same phenomenon happens in the distributions of test positive data points and test negative data points. This illustrates that PET gives more informative representations. (Visualizations on more datasets can be found in Appendix A.2.)

### 4.3.3 Impacts of different retrieval schemes

First, we evaluate the effectiveness of current retrieval by comparing the current retrieval scheme with the random retrieval scheme. The results are summarized in Table 5. One can see that the relevance-based retrieval performs much better than random retrieval. This validates the effectiveness of the relevance-based retrieval and feeding in relevant rows.

We also study the impact of using different retrieval sizes $K$. The results are put in Appendix A.3 due to the page limit. Generally, too few retrieval samples may fail to carry enough information to enhance the representation, while too many retrieved samples will introduce more noise.

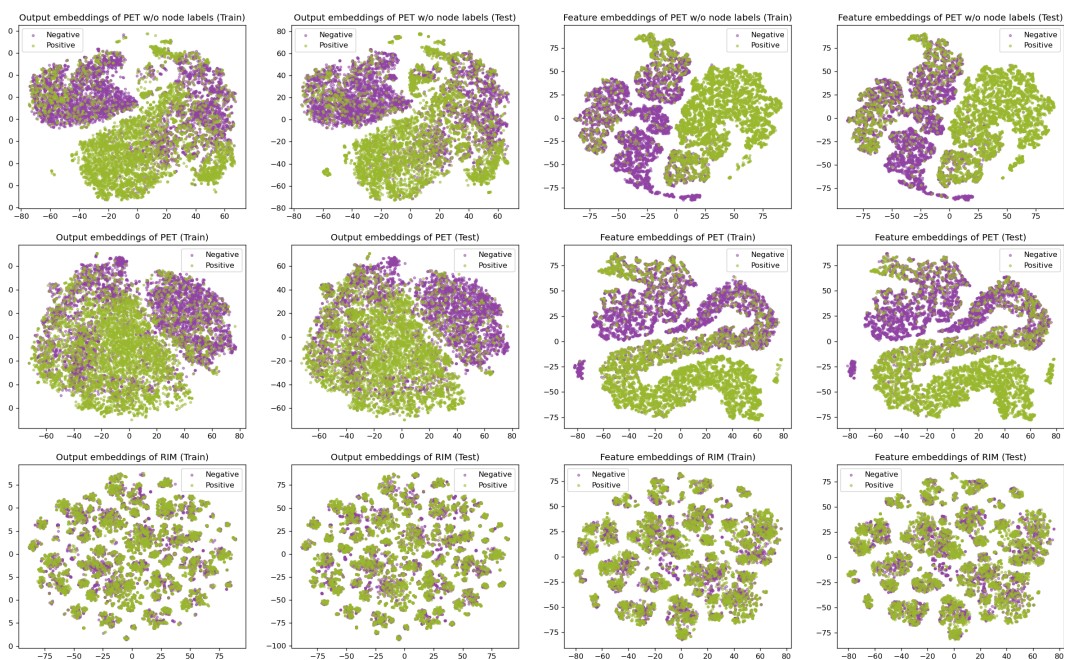

Figure 4: The t-SNE visualization of data and feature embeddings on Tmall.

Table 5: Impacts of the retrieval schemes.

| Models | Tmall | | Taobao | | Alipay | |
|---|---|---|---|---|---|---|
| | AUC | LogLoss | AUC | LogLoss | AUC | LogLoss |
| Random retrieval | 0.8433 | 0.4922 | 0.6544 | 0.6572 | 0.7271 | 0.6120 |
| Relevance retrieval | 0.9324 | 0.3321 | 0.8838 | 0.4162 | 0.8930 | 0.4132 |

## 5   Conclusion

In this paper, we focus on improving the prediction of tabular data, which is essential in many important downstream tasks. Existing methods either take each data instance independently or directly take multiple data instances as input without enhancing the target data instance representation. We propose to construct a retrieval-based hypergraph to model the cross-row and cross-column relations of tabular data, utilizing the propagation on the resulting graph to directly change and enhance the target data instance representations. Concretely, we utilize a relevance retrieval to construct the hyperedges set of the hypergraph, aiming to resort to relevant patterns. Then we design PET, a novel architecture that propagates and enhances the tabular data representations based on the hypergraph for target label prediction. The propagation serves from three aspects: 1) label propagates through common feature values; 2) features get enhanced through locality and high-order product feature interactions are generated through the interactive message passing framework; and 3) labels are used to generate the label-enhanced features. Experiments on two important tabular data prediction tasks validate the superiority of the proposed PET model over strong baselines.

## Acknowledgments and Disclosure of Funding

The Shanghai Jiao Tong University Team is supported by Shanghai Municipal Science and Technology Major Project (2021SHZDZX0102) and National Natural Science Foundation of China (62076161, 62177033). We would also like to thank Wu Wen Jun Honorary Doctoral Scholarship from AI Institute, Shanghai Jiao Tong University.

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
