# OpenReview forum: "Learning Enhanced Representation for Tabular Data via Neighborhood Propagation"
_NeurIPS.cc/2022/Conference — NeurIPS 2022 Accept_

### Official Review · Reviewer_8FC8 · 2022-06-23

**Rating:** 7
**Confidence:** 4
**Soundness:** 3 good
**Presentation:** 3 good
**Contribution:** 2 fair

**Summary:**

The authors propose a novel architecture for learning representations over “tabular data”. The problem setup is highly relevant as learning over tabular data has diverse practical implications in multiple fields. The proposed architecture, named “PET: Propagate and Enhance the Tabular data”, utilizes multiple components at its core - (i) relevance based retrieval to obtain $K$ instances similar to the target data instance;  (ii) message passing (over the star expansion of the generated hypergraph); (iii) label enhanced feature learning; (iv) higher-order feature interaction. To retrieve the relevant instances, the authors utilize the “relevance value” (proposed by Qin et al., 2021), which is akin to BM25 when the data instances are treated as documents and their feature values as the terms, as mentioned by the authors. The resulting set of the original instance along with the retrieved ones is processed as a hypergraph, where each distinct feature value in each individual field forms a node and each data instance constructs a hyperedge. This is further subjected to the star expansion to construct a bipartite graph. Subsequently, few steps of message passing are applied over the graph, with multiple operations involved. The messages are generated by the interaction of the edge and node representations - eq 6. The aggregation of the messages for each node is done by an attention mechanism, attributed to the widespread success of the basic attention module. The node embedding update is then performed via a transformation over the concatenation of previous layer representation with the aggregated messages. The edge embedding update follows a similar process. Finally, the target data node representation is passed through MLP for prediction and the model is learned end-to-end (cross-entropy is used for binary prediction tasks). As emphasized by the authors multiple times over the course of the paper, the proposed architecture aids the feature learning process by inducing the label information into the representations as well as allowing for higher order feature interactions with more rounds of propagation. The empirical analysis is done on 5 majorly used datasets and comparison has been performed against multiple baselines.

**Questions:**

1. In section 4.3.1, the authors have performed two primary ablations by - (i) removing “edge labels” and (ii) removing “node labels”. Do the authors have some results where both “edge” and “node” labels are removed? Essentially, only the “feature interactions” are captured in this setup. This might help validate the efficacy of the structure of the proposed architecture.
2. The authors have utilized “directional” edges in the message passing mechanism. Do the authors have supporting arguments and/or experiments for the case where the graph obtained post the “star” expansion is undirected? In this case, the edge embeddings can be a composition of the two nodes. The composition function can be a simple operation like concatenation, which has been heavily utilized in this work, or something more sophisticated - MLP.
3. Can the authors also describe how they arrived at the hyperparameter values provided in table 8 of the appendix?
4. From figure 7 in the appendix, it is evident that $K=15$ or $K=20$ provide the best results. Is there a specific reason that the authors have used $K=10$ as shown in the hyperparameter table 8 of appendix?


**Limitations:**

As mentioned in section A.1 of the appendix, the constructed graph can be dense when the number of fields is large. Moreover, the graph is created for **each** target instance, which can be computationally expensive. The problems of oversmoothing and oversquashing may occur in such cases. I commend the authors for explicitly stating this. I also believe that the assumption that relevance based retrieval can help realize the homophily may not always hold, as described previously. Although such scenarios can be rectified by utilizing the enhancements proposed in the recent literature focused on learning GNNs under heterophily (such as [1]), I would like to know the authors’ response to this argument. Lastly, I don’t see any serious negative societal impacts of the work.

[1] ​​Graph Neural Networks with Heterophily - Jiong Zhu, Ryan A. Rossi, Anup Rao, Tung Mai, Nedim Lipka, Nesreen K. Ahmed, Danai Koutra


**Strengths And Weaknesses:**

1. The results provided in tables 2, 3 and 4 “strongly” demonstrate the efficacy of the proposed model. The improvements in both - CTR and top-n recommendation tasks are significant.
2. Figure 4 provides interesting visualizations that demonstrate the efficacy of the proposed method PET and inline with the experimental results.
3. While the authors have provided some hyperparameter and sensitivity analyses for label usage (sec 4.3.1), retrieval schemes (sec 4.3.3) and $K$ (appendix A.3), it will be also interesting to see analyses of important attributes like the number of GNN layers and embedding size as well.
4. The authors utilize retrieved neighbors (eq 1 and 2) to obtain the representation of the target data instance under message passing framework. Furthermore, they mention explicitly in section 4.3.3 that the “relevance-based retrieval” can help achieve homophily, and thus improve performance. However, in the cases when relevance based retrieval fetches $K$ instances that are close in the input space, but have different labels, the method can give highly sub-par performance, particularly when the GNN layers are more than $2$. Such scenarios can occur when the output class label is heavily dependent on a few “fields” and the data instances have high similarity in the irrelevant fields (more in count). Can the authors provide some arguments and/or experiments under this setup?
5. The authors have mentioned prior works utilizing GNNs over tabular data in section 2 (lines 77-79), however the comparison in tables 2 and 3 doesn’t contain these baselines. Can the authors elaborate on this? It’s strongly recommended to provide the comparison for the completeness of the work.
6. Tables 6 and 7 of the appendix provide the corresponding standard deviations for the proposed model. I suggest the authors provide the same for the second best method, ie the best performing baseline, for completeness as well.

---

> ### Author Response · Authors · 2022-08-02
> **Response to reviewer 8FCB (Part 3)**
>
> Thanks for all the helpful suggestions. We provide our responses to each concern in the following parts. For your convenience, we give the response to the questions in the sequence the questions appeared.
>
>
> > The authors have utilized “directional” edges in the message passing mechanism. Do the authors have supporting arguments and/or experiments for the case where the graph obtained post the “star” expansion is undirected? In this case, the edge embeddings can be a composition of the two nodes. The composition function can be a simple operation like concatenation, which has been heavily utilized in this work, or something more sophisticated - MLP.
>
> Our graph is indeed an undirected graph (with each undirected edge being represented by two different directional edges between a node pair), but message passing neural network in general still produces directional messages.  And the edge embedding is updated according to the two incident node embeddings. That being said, our choice of message computation is just an example among many other options. The design space could be large and we think your suggestion makes sense as well.
>
> >  Can the authors also describe how they arrived at the hyperparameter values provided in table 8 of the appendix?
>
> The hyperparameters of PET (e.g., embedding size, hidden size, learning rate, l2 norm) were inherited from the RIM (Qin et al.) model. And the inherited hyperparameters worked satisfyingly well. For the baseline models, we used the optimal hyperparameters reported by RIM (Qin et al.) for the non-graph methods. For the graph methods, the learning rate was selected from {1e-4, 3e-4, 5e-4, 1e-3}, and l2_norm was selected from {1e-4, 1e-5, 5e-5}. The results of the baselines were consistent with previous papers.
>
> > From figure 7 in the appendix, it is evident that K=15 or K=20 provide the best results. Is there a specific reason that the authors have used K=10 as shown in@ the hyperparameter table 8 of appendix?
>
> In the baseline RIM paper, they used K=10 on these datasets. Therefore, for fair comparisons, we used the same retrieval size. We can definitely use larger retrieval sizes (K=15, 20) to gain better performance as shown in Figure 7 in the Appendix.
>
> > As mentioned in section A.1 of the appendix, the constructed graph can be dense when the number of fields is large. Moreover, the graph is created for each target instance, which can be computationally expensive. The problems of oversmoothing and oversquashing may occur in such cases. I commend the authors for explicitly stating this.
>
> Thanks for your suggestions.
>
> For oversmoothing, as discussed in our paper, this could be a potential problem, which we leave for future work. Solutions to this situation could be to retrieve and construct the graph over selected fields (e.g., top 10 fields that have the largest mutual information with the labels).
>
> As for the price for constructing graphs, we provide the analysis as follows. Let $K$ indicate the number of retrieved data instances, and let $F$ indicate the number of feature fields. Then, the number of nodes in the graph is no more than $(K+1)(F+1)$, and the number of edges is $2(K+1)F$. Both of them are $O(KF)$, which is linear to the number of fields. In addition, the graph construction for each data instance can be done in parallel for a batch of data. We batch all the resulting graphs for the data instances in the batch to form one single batched graph. Then we can apply GNN on the resulting single batched graph to make predictions parallelly.

---

> > ### Comment · Reviewer_8FC8 · 2022-08-05
> > **Response to the Rebuttal**
> >
> > I commend the authors for answering most of the questions. Although some concerns regarding the problem of *oversmoothing* are left unaddressed, I believe the authors have done sufficient experiments to demonstrate the efficacy of their approach. I am thus increasing the score to **7**.

---

> ### Author Response · Authors · 2022-08-02
> **Response to reviewer 8FCB (Part 2)**
>
> Thanks for the valuable feedback. We provide our responses to each concern in the following parts. For your convenience, we give the response to the questions in the sequence the questions appeared.
>
> > The authors have mentioned prior works utilizing GNNs over tabular data in section 2 (lines 77-79), however the comparison in tables 2 and 3 doesn’t contain these baselines.
>
> Thanks for your suggestion. We compared FATE (Wu et al.) in the paper, which has a similar data-feature bipartite graph and performs GraphSage on the graph. Additionally, we experimented with the recent TabGNN (Guo et al.), which builds multiplex graphs using selected table columns and attentively aggregate over the multiplex graphs. We have added the recent results in Table 2 in the revised paper. For convenience, we also post the results of graph methods used here. We can find that PET still outperforms these graph methods.
>
> | Tmall    | AUC | Logloss | Rel.Impr. |
> | - | - | -- | -- |
> |FATE|0.8553|0.4737|9.01%|
> |TabGNN|0.8945|0.4158|4.24%|
> |PET|0.9324|0.3321|-|0.8838|
>
> |  Taobao   | AUC | Logloss | Rel.Impr |
> | - | - | - | - |
> |FATE|0.6762|0.6497|30.70%|
> |TabGNN|0.7294|0.6173|21.17%|
> |PET|0.8838|0.4162|-|
>
> |  Alipay   | AUC | Logloss | Rel.Impr. |
> | - | - | - | - |
> |FATE|0.7356|0.6199|21.40%|
> |TabGNN|0.8086|0.5849|10.44%|
> |PET|0.8930|0.4132|-|
>
> > I suggest the authors provide the same for the second best method, ie the best performing baseline, for completeness as well.
>
> Thanks for your suggestion. We have added the results in Table 6 and Table 7 in the revised paper. And for your convenience, the results are listed below.
>
> | | |Tmall  | Tmall   | Taobao | Taobao  | Alipay | Alipay  |
> |--|-|-|-|-|-|-|-|
> || |AUC|Logloss|AUC|Logloss|AUC|Logloss|
> |RIM| Mean|0.9138 |0.3804 | 0.8563 |0.4644 |0.8006 |0.5615|
> |RIM|Std.| 0.0023 | 0.0027 | 0.0021 |0.0029 | 0.0026 |0.0031 |
> |PET|Mean| 0.9324 | 0.3321 | 0.8838 | 0.4162 | 0.8930 | 0.4132|
> |PET|Std.| 0.0030 | 0.0036 | 0.0022 | 0.0031 | 0.0024 | 0.0027 |
>
> | (ML-1M) | |HR@1  | HR@5   | HR@10 | NDCG@5  | NDCG@10 | MRR  |
> |--|-|-|-|-|-|-|-|
> |RIM| Mean|0.0645 | 0.2515 |0.4014 | 0.1577 | 0.2059 | 0.1704|
> |RIM|Std.| 0.0032	| 0.0063 |	0.0061|0.0057|0.0048|	0.0043 |
> |PET|Mean| 0.0904 | 0.2889 |0.4404 | 0.1903 | 0.2390 | 0.2006|
> |PET|Std.| 0.0037	| 0.0057 |	0.0073|0.0045|0.0048|	0.0041 |
>
> | (LastFM) | |HR@1  | HR@5   | HR@10 | NDCG@5  | NDCG@10 | MRR  |
> |--|-|-|-|-|-|-|-|
> |RIM| Mean|0.0915 | 0.3468 |0.5780 | 0.2165 | 0.2911 | 0.2210|
> |RIM|Std.| 0.0063|	0.0081	|0.0137 |0.0043 |0.0062|0.0042 |
> |PET|Mean| 0.1149 | 0.3621 |0.6033 | 0.2381 | 0.3156 | 0.2492|
> |PET|Std.| 0.0098|0.0075|0.0074|0.0092|0.0059|0.0088|
>
> We can see that PET consistently outperforms other baselines.
>
> > Do the authors have some results where both “edge” and “node” labels are removed?
>
> Thanks for your suggestions, we did an additional experiment on that by randomly initializing the corresponding node/edge label embeddings. The results have been updated in the revised paper. For your convenience, we also list the table below.
> |  | Tmall  | Tmall   | Taobao | Taobao  | Alipay | Alipay  |
> | - | - | - | - | - | - | - |
> |  | AUC | Logloss | AUC | Logloss | AUC | Logloss |
> | RIM | 0.9120 |0.3769 | 0.8587 | 0.4586 |  0.7845|0.5742 |
> | PET | 0.9279 |0.3387 |0.8762 | 0.4279 | 0.8720 |0.4201 |
> | PET (w/o node labels) | 0.9291 |0.3367 |0.8665 | 0.4465 | 0.8558 |0.4776  |
> | PET (w/o edge labels) | 0.9233 |0.3494 |0.8431 | 0.4847 | 0.8518 |0.4799  |
> | PET (w/o all labels) | 0.9208 | 0.3568 | 0.8416| 0.4815|0.8096| 0.5719  |
>
> One can find that the feature interactions among relevant data instances under the PET framework works too and that feeding in labels helps to enhance the prediction.

---

> ### Author Response · Authors · 2022-08-02
> **Response to reviewer 8FCB (Part 1)**
>
> Thanks for all your helpful feedback and suggestions. We provide our responses to each concern in the following parts. For your convenience, we give the response to the questions in the sequence the questions appeared.
>
> > It will be also interesting to see analyses of important attributes like the number of GNN layers and embedding size as well.
>
> We checked the impacts of the number of GNN layers and the embedding sizes on sampled data, the results are listed below.
> | #Layers (AUC)| Tmall | Taobao | Alipay |
> | - | - | -| - |
> | 2 |0.9236 | 0.8734| 0.8722|
> |3| 0.9279 | 0.8762| 0.8720|
>
> | #Layers (Logloss)| Tmall | Taobao | Alipay |
> | - | -- | -| -- |
> | 2  | 0.3427  |0.4310  | 0.4213|
> | 3  | 0.3387 | 0.4279 | 0.4201 |
>
> | Embedding Size (AUC)| Tmall | Taobao | Alipay |
> | -- | -- | -- | -- |
> | 16 | 0.9279 | 0.8762 | 0.8711 |
> | 32 | 0.9294 | 0.8803| 0.8720 |
>
> | Embedding Size (Logloss)| Tmall | Taobao | Alipay |
> | -- | --| --| --|
> | 16 | 0.3387  | 0.4279 | 0.4237 |
> | 32 | 0.3199 | 0.4211 | 0.4201|
>
> From the results, we can see that the performance of 3-layer PET is better than 2-layer PET. When the number of layers is 3, there are feature interactions among neighboring data instances involved, as we have discussed in Section 3.4.
>
> In addition, we can see that the performance will be a bit better when the embedding size increases. In our paper, we used the embedding sizes adopted by previous papers (16 for Tmall and Taobao, 32 for Alipay).
>
> > Furthermore, they mention explicitly in section 4.3.3 that the “relevance-based retrieval”can help achieve homophily, and thus improve performance. However, In the cases when relevance based retrieval fetches K instances that are close in the input space, but have different labels, the method can give highly sub-par performance, particularly when the GNN layers are more than 2. Such scenarios can occur when the output class label is heavily dependent on a few “fields” and the data instances have high similarity in the irrelevant fields (more in count). Can the authors provide some arguments and/or experiments under this setup?
>
> > I also believe that the assumption that relevance based retrieval can help realize the homophily may not always hold, as described previously. Although such scenarios can be rectified by utilizing the enhancements proposed in the recent literature focused on learning GNNs under heterophily (such as [1]), I would like to know the authors’ response to this argument.
>
> Thanks for the suggestions. It is correct that the assumption that current relevance based retrieval can help realize the homophily may not always hold. A simple solution could be to retrieve over the selected fields (columns) that have large mutual information value with the labels. We can explore this as future work.
>
> To further investigate this issue, we compute the homophily ratios associated with several datasets according to the following equation (the homophily ratio here is computed on the star graph consisting of data instance nodes only):
> $$homophily = \sum_{t}\left(y^t\frac{\sum_{i\in N(t)} y^i}{|N(t)|} +  (1-y^t)(1-\frac{\sum_{i\in N(t)} y^i}{|N(t)|})\right)/\sum_{t}1.$$
> In this way, the homophily ratios for Tmall, Taobao, and Alipay are 0.5566, 0.6111, and 0.5097, respectively. For reference, the commonly used benchmark graph Cora-Full has a homophily ratio of 0.57, as reported in H2GCN:
>
> Zhu et al. "Beyond Homophily in Graph Neural Networks: Current Limitations and Effective Designs." Neurips 2020.
>
> This shows that we obtained homophily graphs on the datasets by doing relevance retrieval. Note also that many graph benchmarks involve much higher homophily ratios (e.g., Citeseer: 0.74, Pubmed: 0.80, Cora: 0.81, same reference), so clearly PET can operate effectively with weaker homophily data (in particular as indicated by Alipay).

---

### Official Review · Reviewer_3NXB · 2022-07-12

**Rating:** 7
**Confidence:** 4
**Soundness:** 3 good
**Presentation:** 3 good
**Contribution:** 3 good

**Summary:**

The authors propose PET, an architecture that Propagates and Enhances the Tabular data representations for label prediction using hypergraphs.  Given tabular data, the novel end-to-end system consists of of several steps, beginning with hypergraph construction based on information retrieval, followed by partitioning of labels and feature into a bipartite graph, then several rounds of message passing to refine high-level interactions between nodes and edges of the bipartite graph, and finally target node embeddings are fed to an MLP for label prediction.  PET is shown to significantly outperform a wide variety of state-of-the-art label prediction methods for two tasks: click-through rate (CTR) prediction and top-n recommendations.  Furthermore, due to the complicated nature of the end-to-end system, an extensive ablation study is conducted to understand the affects different components of the model have on downstream performance.

**Questions:**

What dataset are the T-SNE plots derived from?  Interestingly, PET is learning overarching structure inherent in the data, whereas RIM seems to just learn small clusters.  It would be interesting to see the influence of utilizing edge and node labels on the ability to learn this
structure, i.e., repeating study in Section 4.3.1 to regenerate these T-SNE plots and viewing whether PET (w/o node labels) learns less structure in the data (like RIM T-SNE plots) and the overarching structure becomes more apparent with each edge addition.

Can you comment on the process for choosing PET hyperparameters for the different datasets?  As the hyperparameters differ per dataset, how much computational time does hyperparameter optimization require per dataset?

PET is complicated, can the authors comment on the computational price incurred for the various steps, i.e., hypergraph construction/information retrieval, star expansion, message passing, embedding calculations, MLP training time, and hyperparameter tuning across all these various steps?  How does this compare to RIM?

Question from the previous section: Are the various steps within PEP tractable when the number of features grows on the order of dozens/hundreds, as is commonly encountered in large-scale binary classification datasets?

**Strengths And Weaknesses:**

Originality
Strengths: While PET borrows inspiration from previous methods (i.e, RIM also uses information retrieval to learn inter-row -column high-level interactions), the use of hypergraph neural networks for tabular data prediction is novel and extremely interesting.
------------------
Quality:
Strengths: The approach, methodology, and experiments are all well done.  In particular, the various stages of PET make intuitive sense at drawing out and refining high-level interactions within the data to improve target node prediction.  Furthermore, the ablation study was thorough and illustrative to understand the impact of the hypergraph neural network on downstream performance, as well as visualizing the structured latent space learned in PET compared to the nearest SOTA competitor RIM.

Weaknesses: There are three large concerns given PET.  While the overall model has been shown to perform extremely well, model complexity is high.  While exact runtimes and the necessary compute required to put this model into action for large-scale data are not currently discussed, it does not seem like all researchers will have access to compute resources necessary to train and deploy such a model.

The second concern is related to the first; how does runtime scale with the number of features considered?  While the datasets utilized in the paper contain large numbers of samples, they contain very small numbers of fields.  Are the various steps within PEP tractable when the number of features grows on the order of hundreds, as is commonly encountered in large-scale binary classification datasets?

Finally, it is not clear how computationally expensive hyperparameter optimization of PET is, nor how often this is necessary (each presented dataset has a different configuration of hyperparameters).

------------------
Clarity:
Strengths: The paper is very well written.

Weaknesses: A necessasry discussion of hyperparameter tuning for the various methods evaluated is currently lacking in the paper.  In particular, for the competitors in Table 2 and 3, the choice of hyperparameters is not discussed.

------------------
Significance:
Strengths: GNNs are an extremely active area of research right now.  Coupled with the impressive performance demonstrated for two important tasks, PET has the potential to significantly influence current work looking to similarly leverage GNNs to capture high-level latent interactions among features for different prediction tasks.

---

> ### Author Response · Authors · 2022-08-02
> **Response to reviewer 3NXB (Part 2)**
>
> Thanks for all your valuable review and feedback. This part we give the response to the remaining questions on model tuning and the PET (w/o node labels) T-SNE plots. The reviewer's main concern on the computation cost can be found in Part 1.
>
> > It is not clear how computationally expensive hyperparameter optimization of PET is, nor how often this is necessary.
> > Can you comment on the process for choosing PET hyperparameters for the different datasets? As the hyperparameters differ per dataset, how much computational time does hyperparameter optimization require per dataset?
>
> We did not spend effort on tuning PET. Instead, the hyperparameters were directly inherited from the RIM (Qin et al.) model, which worked satisfyingly well.
>
> > A necessary discussion of hyperparameter tuning for the various methods evaluated is currently lacking in the paper. In particular, for the competitors in Table 2 and 3, the choice of hyperparameters is not discussed.
>
> For the baselines, we followed previous works. Concretely, we used the optimal hyperparamters reported by RIM (Qin et al.) for the non-graph methods. For the graph methods, the learning rate was selected from {1e-4, 3e-4, 5e-4, 1e-3}, and l2_norm was selected from {1e-4, 1e-5, 5e-5}.
>
> > repeating study in Section 4.3.1 to regenerate these T-SNE plots and viewing whether PET (w/o node labels) learns less structure in the data (like RIM T-SNE plots) and the overarching structure becomes more apparent with each edge addition.
>
> Thanks for your suggestion. We did T-SNE plots with PET (w/o node labels) on the datasets. The results have been updated in Figure 4, Figure 5, and Figure 6 in the revised paper. From the visualization results, we can see that PET w/o node labels still learns a better structure than RIM. And PET learns better representations than PET w/o node labels.

---

> > ### Comment · Reviewer_3NXB · 2022-08-09
> > **Response to authors**
> >
> > I thanks the authors for their response.
> >
> > With regards to computational complexity, the rigorous answer and accompanying numbers are appreciated.  Additionally, it would be helpful to have some concrete runtimes for the various steps on the different datasets.
> >
> > Including the authors' response about hyperparameters (across the various methods) in the main paper would also help prevent any confusion for readers parsing the included results.

---

> > > ### Author Response · Authors · 2022-08-09
> > > **Response to reviewer 3NXB**
> > >
> > > Thanks for your valuable feedback, which we appreciate a lot.
> > > We have added the descriptions of the hyperparameters and the complexity analysis in the revised paper.
> > >
> > > In addition, for your information, we offer the exact runtime for each stage as below.
> > > *All the experiments were on one Tesla T4 instance. The runtime was calculated on the Alipay dataset.
> > >
> > > - Retrieval time: (Both RIM and PET cost the same.)
> > >     - Cost for building elasticsearch index: （**This could be done offline and only once.**）Inserting 1,000,000 data samples costs 00:00:49. For Alipay, we inserted a total of 82,090,360 rows, which cost 1:07:30.
> > >     - Cost for search: We searched 10 neighbors for each target row. Searching neighbors for 264,843 target rows cost 00:05:41.
> > >
> > > - Graph construction and model forward time: The batch size is set to 100 (as used in the paper). For RIM, the total time of model forward for 264,843 rows  is 00:03:14. And for PET, the total time of constructing graphs and model forward for 264,843 rows is 00:03:55.
> > >
> > > *The time is just for reference. The current implementation of PET could be further optimized.

---

> ### Author Response · Authors · 2022-08-02
> **Response to reviewer 3NXB (Part 1)**
>
> Thanks for your helpful review and summary. The reviewer's main concern lies in the computational cost. This part we provide our response to it.  Responses to other questions can be found in Part 2.
>
> > Can the authors comment on the computational price incurred for the various steps, i.e., hypergraph construction/information retrieval, star expansion, message passing, embedding calculations, MLP training time, and hyperparameter tuning across all these various steps? How does this compare to RIM?
>
> > Are the various steps within PEP tractable when the number of features grows on the order of dozens/hundreds?
>
> Let $K$ indicate the number of retrieved data instances, $F$ indicate the number of feature fields, $N$ be the number of unique features, and $d$ be the embedding size.
>
> For the retrieval stage, both PET and RIM cost the same. As discussed in the RIM paper, the average length of the posting lists in the inverted index is $\frac{|SearchPool|}{N}$. The total time complexity of retrieval is $O(F\frac{|SearchPool|}{N})$. As for constructing the inverted index, it could be done offline and only once. The complexity of it is $O(𝐹|SearchPool|)$.
>
> The main computation of PET lies in the message passing part. However, the computation of message passing is linear to the number of feature fields (and is independent of the number of unique features). In addition, graph construction and message passing can be performed highly in parallel. These observations make PET scalable. We provide concrete analysis and comparisons with RIM below.
>
> **Complexity for the attention aggregation and mutual interaction of RIM:** The attention of RIM is computed over the retrieved instances, with each data instance having a representation of size $Fd$. The attention aggregation of RIM takes $O(F^2d^2 + KFd)$. The mutual interaction among the target feature, aggregated neighboring feature, and aggregated neighboring label of RIM takes $O(F^2d)$. Therefore, the total computation of aggregation and interaction for RIM is $O(F^2d^2 + KFd)$.
>
> **Complexity for the message passing of PET:** The number of nodes in the graph for a data instance is no more than $(K+1)(F+1)$, and the number of edges is $2(K+1)F$. Both of them are $O(KF)$. The time complexity of single-layer message passing of PET is $O(KFd^2)$. This makes PET scalable as the complexity is linear with respect to the number of feature fields $F$. In addition, graph construction and message passing can be performed highly in parallel:
> - The graph construction for each data instance can be done in parallel for a batch of data. We batch all the resulting graphs for the data instances in the batch to form one single batched graph. Then we can apply GNN on the resulting single batched graph to make predictions parallelly.
> - For the GNN, the operation of the attention computation can be parallelized across all edges, and the computation of representations can be parallelized across all nodes and edges.
>
> **Complexity for the MLP computation for RIM and PET:** The input vector size of the MLP for RIM is $O(Fd)$. The MLP computation of RIM is $O(F^2d^2)$. While for PET, the input size of the the MLP is $d$. Therefore, the MLP computation of PET is $O(d^2)$, which is independent of the number of feature fields.
>
> In addition, the number of model parameters of PET is lighter than that of RIM. Both other methods (e.g., DeepFM, RIM, etc.) and PET have embedding tables of the same size $Nd$. The main differences of the parameter numbers lie in the linear layers. For PET, the parameter count of linear layers is $O(d^2)$. While for RIM, the parameter count of linear layers is $O(F^2d^2)$.
>
> For further information, we provide the concrete numbers of model parameters, GPU occupation, and the statistics of the datasets.
>
> | Dataset | Retrieval Size ($K$)    | #Fields ($F$) | #Features ($N$) | #Rows       |
> | ------- | --- | ------------- | --------------- | ----------- |
> | Tmall   |  10   | 9             | 1,529,676       | 54,925,331  |
> | Taobao  |  10   | 4             | 5,159,462       | 100,150,807 |
> | Alipay  |  10   | 6             | 3,327,205       | 35,179,371  |
>
> | Model    | #Params (Tmall) | #Params (Taobao) | #Params  (Alipay) |
> | ---- | ----- | ---- | ----- |
> | RIM                   |       24,589,225              |    82,607,913                  | 106,594,049          |
> | PET (#GNN layers = 2) | 24,500,393          | 82,576,969           | 106,515,033          |
> | PET (#GNN layers = 3) | 24,503,033          | 82,579,609           | 106,525,433          |
>
> | Model                 | Max GPU Occupied (Tmall) | Max GPU Occupied (Taobao) | Max GPU Occupied (Alipay) |
> | ----- | ----- | ---- | -- |
> | RIM  | 2,070 M| 3,680 M   |4,346 M |
> | PET (#GNN layers = 2) |1,761 M | 3,375 M | 4,057 M |
> | PET (#GNN layers = 3) |   1,849 M       |   3,481 M    |   4,061M  |

---

### Official Review · Reviewer_9mtT · 2022-07-15

**Rating:** 5
**Confidence:** 4
**Soundness:** 3 good
**Presentation:** 3 good
**Contribution:** 2 fair

**Summary:**

The authors propose a graph approach that connects an item to classify and its features with relevant items via the common values. This graph is used to propagate feature and label information and then classify the item. Comparisons are made in multiple datasets for two tasks (recommendation and classification) with several state-of-the-art approaches. The results show that the proposed approach can improve the performance on the specific metrics.

**Questions:**

One question I have is about the relevant item extraction. This works well for binary or categorical features but did you try with real-valued features?




**Ethics Review Area:**

["I don’t know"]

**Limitations:**

As I mentioned previously the retrieval process as defined seems to limit the applicability of the approach for the kind of features the authors consider.

**Strengths And Weaknesses:**

This is an interesting approach and based on the results it seems to improve upon various state-of-the-art models. It is described quite well, even though I would like to see a bit more detailed description of the implementation. Some algorithm pseudo-code would help. Experiments are good with several models for comparison. The results are also good and not negligible I would say.

Something that is missing in the paper is the detailed description of how the baselines were tuned. For example, DeepFM is quite a strong approach but one need to tune properly from my experience. So, I am not sure if the difference is due to the approach that better captures and propagates the relevant information or just note well tuned approaches.

The authors could also have added graph approaches like for example GraphSage or other state-of-the-art ones.

Also, the retrieval of relevant instances is thoroughly studied. This has a big impact I imagine on the learning process. I would like to see here different scenarios and more analysis.

---

> ### Author Response · Authors · 2022-08-02
> **Response to reviewer 9mtT (Part 1)**
>
> Thanks for your helpful review and summary. We give our responses to your concerns one by one.
> This part gives the detailed algorithm and experimental setting. The reviewer's question on the retrieval is given in the next part.
>
> > It is described quite well, even though I would like to see a bit more detailed description of the implementation. Some algorithm pseudo-code would help.
>
> We divided the implementation into three stages and offered a detailed description of each stage from the perspective of one target instance (parallel batch prediction can be done by simple batching): 1) retrieval-based hypergraph construction as described in Section 3.1, 2) message passing as described in Section 3.2, and 3) target node embeddings being fed into the MLP as described in Section 3.3. A pseudo-code of PET has been added in the Appendix A.1 in the revised paper. For your convenience, we also offer the code in plain text here:
>
> **Input**: Target data instance (row) $X^t=\{x_i^t|i=1,\dots,F\}$,  Retrieval Pool $D_{ret}$.
>
> Retrieve $K$ most relevant data instances for $X^t$ from the retrieval pool $D_{ret}$ to obtain $\{X^{r_1}, \cdots, X^{r_K}\}$ according to Equation (1).
>
> Construct a hypergraph $G=(V_D,V_F,E)$ from hyperedges $\{X^t, X^{r_1}, \cdots, X^{r_K}\}$.
>
> Initialize the node and edge embeddings according to Equations (3), (4), and (5).
>
> **for** $l\in\{1,\dots, L\}$ **do**
>
> $\qquad$**for** $j\in V_D \cup V_F$ **do**
>
> $\qquad$$\qquad$$n_j^{(l)}=AttenAGG^{(l)}\left(\{(e_{ij}^{(l-1)}\odot h_i^{(l-1)})\|h_i^{(l-1)} | \forall i\in N(j)\}\right)$.
>
> $\qquad$$\qquad$$h_j^{(l)} = \sigma( W_N^{(l)}(h_j^{(l-1)}\|n_j^{(l)}))$.
>
> $\qquad$**end**
>
> $\qquad$**for** $(i,j)\in E$ **do**
>
> $\qquad$$\qquad$$e_{ij}^{(l)} = \sigma( W_E^{(l)}(h_i^{(l)}\|h_j^{(l)}\| e_{ij}^{(l-1)}))$.
>
> $\qquad$**end**
>
> **end**
>
> $\hat{y_t}\leftarrow MLP(h_t^{(L)})$.
>
> > Something that is missing in the paper is the detailed description of how the baselines were tuned.
> > I am not sure if the difference is due to the approach that better captures and propagates the relevant information or just note well tuned approaches.
>
> For the baseline tuning, we followed previous work. Concretely, we used the optimal hyperparameters reported by RIM (Qin et al.) for the non-graph methods. For the graph methods, the learning rate was selected from {1e-4, 3e-4, 5e-4, 1e-3}, and l2_norm was selected from {1e-4, 1e-5, 5e-5}. The performances are consistent with the previous work.
>
> > The authors could also have added graph approaches like for example GraphSage or other state-of-the-art ones.
>
> Thanks for your suggestion. We compared against FATE (Wu et al.) in the paper, which has a similar data-feature bipartite graph and performs GraphSage on the graph. Additionally, we experimented with the recent TabGNN (Guo et al.), which builds multiplex graphs using selected table columns and attentively aggregate over the multiplex graphs. We have added the recent results in Table 2 in the revised paper. For convenience, we also post the results of graph methods used here. We can find that PET still outperforms these graph methods.
>
>
> | Tmall    | AUC | Logloss | Rel.Impr. |
> | --- | ----------- | --------------- | ----------------- |
> |FATE|0.8553|0.4737|9.01%|
> |TabGNN|0.8945|0.4158|4.24%|
> |PET|0.9324|0.3321|-|0.8838|
>
> |  Taobao   | AUC | Logloss | Rel.Impr |
> | --- | ----------- | --------------- | ----------------- |
> |FATE|0.6762|0.6497|30.70%|
> |TabGNN|0.7294|0.6173|21.17%|
> |PET|0.8838|0.4162|-|
>
> |  Alipay   | AUC | Logloss | Rel.Impr. |
> | --- | ----------- | --------------- | ----------------- |
> |FATE|0.7356|0.6199|21.40%|
> |TabGNN|0.8086|0.5849|10.44%|
> |PET|0.8930|0.4132|-|

---

> ### Author Response · Authors · 2022-08-02
> **Response to reviewer 9mtT (Part 2)**
>
> Thanks for all your valuable feedback. We give our response to the reviewer's concern on the retrieval and the applicability of the approach here. Response on other questions can be found in Part 1.
>
> > Also, the retrieval of relevant instances is thoroughly studied. This has a big impact I imagine on the learning process. I would like to see here different scenarios and more analysis.
>
> Yes, the design space of PET could be large and flexible. Generally, we model the set relationship within a row with a hyperedge. And we use retrieval to capture the cross-row dependencies. The options for the retrieval algorithm could be flexible, like hard search using dominant fields (e.g., same user ID).
>
> In the paper, we compared with random retrieval in Table 5 and studied the impact of different retrieval sizes in Figure 7. From the results, we can see that the relevance retrieval helps to capture cross-row dependencies. In addition, more retrieved instances can contain more auxiliary information and give better results, but too many retrieved instances may introduce noise.
>
> To compare different retrieval strategies, we did an additional experiment with hard retrieval on sampled data. The results are listed below.
>
> | Retrieval Algorithm (AUC) | Tmall  | Taobao |   Alipay  |
> | ------------------------- | ------ | ------- | --- |
> | Random  retrieval         | 0.8441   | 0.6509    |  0.7137   |
> | Hard retrieval            | 0.9392   | 0.8603    |  0.8667   |
> | Relevance retrieval (BM25)      | 0.9279 | 0.8762    |  0.8720   |
>
> | Retrieval Algorithm (Logloss) | Tmall  | Taobao |   Alipay  |
> | ------------------------- | ------ | ------- | --- |
> | Random  retrieval         | 0.4901   | 0.6581    |   0.6223  |
> | Hard retrieval            | 0.3211   | 0.4479    |  0.4238   |
> | Relevance retrieval   (BM25)    | 0.3387 | 0.4279    |  0.4201   |
>
>
> There are also more use cases of relevant instance retrieval. Some click-through rate (CTR) models have retrieved over user history to capture user behavior patterns, e.g., UBR (Qin et al.). Also, the relevant retrieval can be seen as a good way to obtain the context, e.g, we can use the relevant instances to perform pretraining.
>
> > One question I have is about the relevant item extraction. This works well for binary or categorical features but did you try with real-valued features?
> > As I mentioned previously the retrieval process as defined seems to limit the applicability of the approach for the kind of features the authors consider.
>
> Actually, there already are continuous features in the datasets we used (e.g., age, timestamps, etc.) for the experiments. We performed discretization for the continuous features, which is a widely-used method in current tabular models like AutoDis:
>
> Guo, Huifeng, et al. "An embedding learning framework for numerical features in ctr prediction." In Proceedings of the 27th ACM SIGKDD Conference on Knowledge Discovery & Data Mining, pp. 2910-2918. 2021.

---

### Meta-Review · Area_Chair_qezP · 2022-08-25

**Recommendation:** Accept
**Confidence:** Less certain

**Metareview:**

This paper proposes (PET), an approach to classifying rows in tabular data using retrieval methods and hypergraph neural networks to make predictions. The key ideas are
- use information retrieval techniques to find similar rows to each row that needs to be labeled.
- connect the similar rows in a hypergraph structure.
- learn a representation over the hypergraph structure with graph neural networks.

Experiments show that PET can singificantly outperform multiple state of the art methods on two tasks. Ablations also validate the design and each component of PET. During the review process, the authors added additional experiments addressing many of the reviewers open concerns.

The reviewers agreed that the paper is very well written, presents a significantly useful method, and that while PET builds on pieces that have been developed separately, it combines them in an interesting way. Reviewers also felt that the work is likely to be of interest to the wider graph neural network community and has the potential to influence future work.

**Award:**

No

---

### Decision · Program_Chairs · 2022-09-14

Accept